# MYB-Mediated Regulation of Anthocyanin Biosynthesis

**DOI:** 10.3390/ijms22063103

**Published:** 2021-03-18

**Authors:** Huiling Yan, Xiaona Pei, Heng Zhang, Xiang Li, Xinxin Zhang, Minghui Zhao, Vincent L. Chiang, Ronald Ross Sederoff, Xiyang Zhao

**Affiliations:** 1State Key Laboratory of Tree Genetics and Breeding, Northeast Forestry University, Harbin 150040, China; huiling682937@163.com (H.Y.); zhangheng815@nefu.edu.cn (H.Z.); lx2016bjfu@163.com (X.L.); zhangxinxin@nefu.edu.cn (X.Z.); zhaominghui66@163.com (M.Z.); vchiang@ncsu.edu (V.L.C.); 2Harbin Research Institute of Forestry Machinery, State Administration of Forestry and Grassland, Harbin 150086, China; xiaonapei2020@163.com; 3Research Center of Cold Temperate Forestry, CAF, Harbin 150086, China; 4Forest Biotechnology Group, Department of Forestry and Environmental Resources, North Carolina State University, Raleigh, NC 27695, USA; ron_sederoff@ncsu.edu

**Keywords:** MYB, transcription factor, anthocyanin, positive regulation, negative regulation, environment, MBW complexes

## Abstract

Anthocyanins are natural water-soluble pigments that are important in plants because they endow a variety of colors to vegetative tissues and reproductive plant organs, mainly ranging from red to purple and blue. The colors regulated by anthocyanins give plants different visual effects through different biosynthetic pathways that provide pigmentation for flowers, fruits and seeds to attract pollinators and seed dispersers. The biosynthesis of anthocyanins is genetically determined by structural and regulatory genes. MYB (*v-myb* avian myeloblastosis viral oncogene homolog) proteins are important transcriptional regulators that play important roles in the regulation of plant secondary metabolism. MYB transcription factors (TFs) occupy a dominant position in the regulatory network of anthocyanin biosynthesis. The TF conserved binding motifs can be combined with other TFs to regulate the enrichment and sedimentation of anthocyanins. In this study, the regulation of anthocyanin biosynthetic mechanisms of MYB-TFs are discussed. The role of the environment in the control of the anthocyanin biosynthesis network is summarized, the complex formation of anthocyanins and the mechanism of environment-induced anthocyanin synthesis are analyzed. Some prospects for MYB-TF to modulate the comprehensive regulation of anthocyanins are put forward, to provide a more relevant basis for further research in this field, and to guide the directed genetic modification of anthocyanins for the improvement of crops for food quality, nutrition and human health.

## 1. Introduction

Anthocyanins are abundant natural water-soluble plant pigments in their glycosylated form [1,2]. Anthocyanin biosynthesis in plants is a part of the phenylpropanoid pathway, which produces flavonoids through flavonoid branches. The content and composition of anthocyanins results in a series of orange, red, purple and blue colors in vegetative and reproductive plant organs. Red anthocyanins most likely evolved along with mosses as plants became adapted to the land about 450 million years ago (mya). The blue pigments evolved later, about 300 mya along with the gymnosperms [1]. Anthocyanins or their orthologs are have been found in some mosses [3], ferns [4] and some fungi [5], but not in algae [1]. Dating back to the early 20th century, anthocyanins have been studied by humans [6]. Anthocyanins are now found in 73 taxa and 27 families of plants [6]. The anthocyanins content varies greatly depending on the variety, species, season, climate, 87and stage of plant development. In higher plants, the rich colors of the anthocyanins attract animals for pollination and seed transmission but may also act in the wild as warning coloration [7,8]. Anthocyanins also regulate the negative effects of the external environment as protection against cold or drought [9,10]. Anthocyanin-rich plants are used in landscape gardens and enrich the natural landscape. There are anthocyanins in leaves, stems, roots, flowers and fruits of diverse plant species such as grape (*Vitis vinifera*), parsley (*Petroselinum crispum*), eggplant (*Solanum melongena*), apple (*Malus domestica*), strawberry (*Fragaria vesca*), mulberry (*Morus alba*), and petunia (*Petunia hybrida*) [11,12,13,14,15,16,17]. Anthocyanins typically accumulate in the vacuoles and their biosynthesis is mediated by many enzymes in the phenylpropanoid metabolic pathways. Anthocyanin synthesis are mainly controlled by two types of genes, one is the anthocyanin biosynthesis structural gene, which can encode the enzyme of anthocyanin biosynthesis pathway, and the other is the regulatory gene, which has three types of transcription factors (TFs): MYB (*v-myb* avian myeloblastosis viral oncogene homolog) protein, bHLH (basic helix-loop-helix) protein, WD40 (WD-40 has a scaffolding function) protein [18]. In monocotyledon (maize, *Zea mays* [19], rice, *Oryza sativa* [20]), MYB-TFs regulate anthocyanin biosynthesis enzymes (such as *CHS*, *F3H* and *DFR*, *LDOX*, *BAN/ANR*, *UFGT*) together with other TFs [21]. In dicotyledon (*Arabidopsis thaliana* [22], apple, *M. domestica* [12]), anthocyanin synthases are divided into two classes and the TFs are different [23]. The *CHS*, *CHI*, *F3H* and *F3‘H* genes are the early biosynthetic genes (EBGs) in the anthocyanin pathway. These genes bind directly and regulate by MYB-TFs [23]. *DFR*, *LDOX*, *BAN*/*ANR*, and *UFGT*, are the late biosynthesis genes (LBGs) [24,25], are regulated by the MBW-TF ternary protein complex of MYB-bHLH-WD40 that controls the MBW complex and the downstream accumulation of anthocyanins [26,27,28,29]. The three most common anthocyanins are: delphinidin-3-glycoside (blue/purple) cyanidin-3-glucoside (brick red/magenta) and pelargonidin-3-glycoside (orange/red). Other important anthocyanins are: malvidin, peonidin and leucocyanidin [30,31]. For many anthocyanins, the color depends on the pH. Cyanidin is red/violet at pH 7-8 changing to blue at higher pH. Delphinidin at high pH is a strong blue color that is common in flowers. Peonidin is cherry red at low pH but deep blue at pH8 [32]. Meanwhile, anthocyanins are considered beneficial for human health; presumably as antioxidants that reduce the abundance of free radicals (reactive oxygen species (ROS)), which may de-lay aging, ameliorate cardiovascular and neurogenerative disease, as well as modulating gut microbiota [7,33], however it has been argued that direct evidence of the benefits of dietary supplements are lacking [34].

In plants, MYB transcription factors (TFs) are one of the most important classes of transcriptional regulators in the plant metabolic network, controlling secondary metabolism, development, signal transduction and resistance to biotic and abiotic stresses [35]. The MYB-TF family is one of the largest in plants. Anthocyanin are secondary metabolites of flavonoids modify with are a subclass of flavonoids, the anthocyanin biosynthesis as well as that of other phenylpropanoids are regulated by MYB-TFs. For example, *MdMYB10a* of apple (*M. domestica*) when overexpressed in Nicotiana tabacum upregulates the biosynthesis of anthocyanins [36], through an increase in the expression levels of structural genes in anthocyanin biosynthesis, thereby activating a hierarchical cascade of the downstream regulatory and structural genes for anthocyanins. Similar results were reported for *LdMYB6* from lily (*Lilium davidiivar*), peach *PpMYB7* (*Prunus persica*), *Arabidopsis AtMYB75* and *AtMYB90* and eggplant *SmMYB113* (*Solanium melongena*) [37,38,39,40]. In a transient transactivation experiment with strawberry (*F. vesca*), *FvMYB10* co-expression with *FvbHLH33* strongly activated the *AtDFR*, *FvDFR* and *FvUFGT* promoters of the structural genes of anthocyanin biosynthesis. Knockouts of *FvMYB10* or *FVbHLH33* significantly reduced the activity of the *AtDFR* promoter in tobacco [41]. MYB proteins contain an imperfect repeat sequence ® which has varying forms in the conserved DNA domain. MYB-TFs are classified into four types based on the number of repeats (1R, 2R, 3R, 4R) and the variation in the sequences of the repeats. The MYB that is the major activator of anthocyanin biosynthesis comes from the two-repeat class (2R) containing an R2 and an R3 repeat (R2R3-MYB) (Figure 1). Variation in relative abundance and specificity of anthocyanins represents one of the most common evolutionary changes in flower color. MYB-TFs may also act as antagonistic repression of anthocyanin enrichment [42]. The first suppressor of anthocyanins identified was the *AmMYB308* gene of snapdragon (*Antirrhinum majus*) [43]. Subsequently, repressors of anthocyanin synthesis were found in strawberry *FaMYB1* (*Fragaria* × *ananassa*), *Arabidopsis AtMYB4*, petunia *PhMYBx* (*Petunia hybrida*), peach *PpMYB17-20* (*P. persica*), popular *PtrMYB182* (*Populus tremula* × *Populus*), and apple *MdMYB16* (*M. domestica*) [11,44,45,46,47,48]. The anthocyanins biosynthesis pathway in the plant concerned is inhibited, resulting in a decrease in the level of anthocyanin in vivo when they were expressed. The repressors identified are in the subgroups of R2R3-MYBs and R3-MYBs. When the putative suppressors were expressed in tobacco, the structural genes of the anthocyanin biosynthesis were inhibited, reducing the abundance of anthocyanin pigments. Environmental factors also affect anthocyanin accumulation [27,49]. In *A. thaliana*, *AtPAP1* is regulated by high light and sucrose levels [50,51], which activate anthocyanin synthesis. During storage of kiwifruit (*Actinidia chinensis*), low temperature results in the upregulation of *AcMYBA1-1* and *AcMYB5-1* and induces anthocyanin biosynthesis [51].

Predictive computational analysis has led to new uses for MYB-TFs by changing the amino acid sequence of the MYB domain to expand its regulatory ability. For example, the anthocyanin produced by the *MdMYB10* gene can be used as a visible selection marker for the transformation of apples, strawberries and potatoes to replace antibiotic selection for kanamycin resistance [54] by changing the amino acids of the R3 domain of *Populus tomentosa MYB6* to generate a *MYB6-R3m* mutant [55]. It is of significant value to understand more about the regulation of anthocyanins by MYB-TFs because of the commercial value of plant color in agriculture. Here we summarize the recent studies of the regulation of anthocyanin synthesis by MYB-TFs and analyze the specificity of positive and negative regulation of MYBs to lay a foundation for more comprehensive work in the future on the functional mechanisms and evolution of this important class of transcriptional regulators. With sufficient information on the mechanism of MYB-target DNA interactions, designing new MYB based regulators could become an important tool to design new specific interactions for synthetic metabolism and biology.

## 2. Structure and Evolution of MYB

A MYB-TF (*v-myb*) was initially identified as an oncogene in an avian myeloblastosis retrovirus (AMV)) that produced leukemia in chicken embryos [56,57]. Initially, Klempnauer et al. found the sequence differences in the *v-myb* gene and its homolog *c-myb*, leading to inferred effects on their protein structures [56]. Paz-Ares et al. (1988) discovered a MYB-TF as the *C1* gene of wild-type maize (*Z. mays*) and described it as the first regulatory transcription factor identified in plants [58]. Since then, 63,028 MYB-TF genes have been identified and officially registered in the National Center for Biological Information (NCBI- GeneBank).

A hydrophobic amino acid core consisting of a series of tryptophan residues specifically binding to the main groove of the DNA double helix, form a conserved MYB DNA binding domain. The MYB domain usually binds to a specific DNA sequence (C/TAACG/TG). The three conserved tryptophan residues are separated by 18 to 19 amino acids form a secondary structure [35]. The MYB domain contains three incomplete repeats (R) with approximately 52 amino acid residues at the N-terminus of the protein (Figure 1). Each repeat encodes three helices. The second and third helix form a helix-turn-helix segment that binds to the major groove of the target DNA [52,59,60]. Once bound, transcription can be initiated to activate a hierarchical regulatory network or the MYB-TF may interact with other TFs to regulate the expression of target genes. Animal MYB proteins diverged from a common ancestor, while the MYB proteins in plants are a polyphyletic group related only to a helix-turn-helix ‘‘MYB-box’’ DNA-binding motif [59]. Based on the number of MYB repeats (R), the (1R, 2R, 3R, 4R) MYBs can be divided into four classes. An evolutionary relationship of the four MYB repeat types has been proposed (Figure 1) [53]. The first category has a repetitive sequence contained in a single protein, often with homologous structural domain proteins in combination with DNA, namely, R1/2-MYB. The R1/R2 repeat is thought to be an antecedent that gave rise to the R1 and R2 repetitive sequences by duplication and divergence. The R3-MYB repeat is found as repressor. *StMYB1* (from potato) was the first MYB protein found to contain only one MYB domain structure identified in plants [33,52]. The second class consists of two R repeats (R1/2R3-MYB and R2R3-MYB), and both R2 and R3 are required for sequence-specific binding, the C-terminal helix of each repeat being the recognition helix for DNA binding [53]. It has been proposed that R2R3-MYB evolved from an R1R2R3-MYB after the loss of the R1 repeat sequence [53,60,61]. Therefore, members of the superfamily of MYB proteins should be viewed as related principally by their ability to bind DNA, rather than on the basis of their physiological functions [53]. Due to the distinct specific binding ability of different MYB proteins and the differences in target gene recognition sites, there is great flexibility of R2R3-MYB binding sites, leading to a wide range of regulatory roles and functions throughout the process of plant growth, metabolism, development, behavior, or adaptation. The third type of MYB is the three repeats MYB, which is a member of the R1R2R3-MYB class. R1 is more conserved than R2 and R3 in type 3R-MYBs, so it is suggested that 3R-MYBs evolved into R1/2R3-MYBs [60]. Current reports indicate that its main function is to participate in cell cycle and protein control and cell differentiation [59,61]. The fourth class, 4R MYBs, which members contain four R1/R2-like repeats (R1R2R2R1/2) [52]. The 4R-MYB has the longest peptide chain (1350 amino acids) consisting of 38,711 bases in flax [62]. The functional expression of 4R-MYB is still under investigation, and its regulatory function is unclear. Additional variation that is found within types of repeats provides further classification into subtribes.

## 3. Anthocyanin Biosynthesis Pathway

Anthocyanins are highly abundant and exceptionally diverse in structure and function among plant phenolic metabolites [63]. Walheldale (1911) explored the process of anthocyanin formation through the phenylpropionic acid and flavonoid pathways [6]. The basic anthocyanin structure consists of two benzene rings and three connecting carbon atoms (C6-C3-C6) [64]. The main metabolic pathways of anthocyanins are mapped in Figure 2 [23,26,65]. Anthocyanin biosynthesis is mediated and regulated by a variety of enzymes to regulate synthesis. The EBGs (*CHS*, *CHI*, *F3H* and *F3‘H*) and LBGs (*DFR*, *LDOX*, *BAN*/*ANR*, and *UFGT*) were initially categorized accordingly to their coordinate expression in response to environmental cues, such as light, at distinct developmental stages, in a species-dependent manner [23,26,27,28,29]. The biosynthesis does not follow a strictly linear pathway but is better described as a metabolic grid, with many branches and alternative metabolic routes. The biosynthesis begins with the deamination of phenylalanine by phenylalanine ammonia lyase (PAL) to cinnamic acid, followed by the hydroxylation of cinnamic acid by cinnamate 4-hydroxylase (C4H) to 4-coumaric acid. 4-coumaric acids activated by 4-cinnamate-CoA ligase (4CL) to 4-coumaryl CoA. Chalcone synthase (CHS) is the key reaction involved in flavonoid biosynthesis [66]. In plants 4-coumaryl CoA is condensed with 3 molecules of malonyl-CoA to form a new aromatic ring system of polytetraketone [67]. As one of the important branches of flavonoid and anthocyanin synthesis, flavonone 3-hydroxylase (F3H), with a 4,5,7-trihydroxyflavanone-catalyzed reaction produces dihydroflavonol [68]. Then, dihydroflavonol is catalyzed by dihydrofavonol 4-reductase (DFR) to synthesize colorless anthocyanins, which are the basic skeletons of flavonoids and anthocyinodins, which are the basic skeletons of flavonoids and anthocyanins [69]. Finally, the colorless dihydroflavonols are converted into colored anthocyanins by anthocyanidin synthase (ANS) [67] and flavonoid 3-O-glucosyltransferase (UFGT), which transforms the colorless dihydrogen flavonols into colored anthocyanins [70]. Subsequently, after UDP -glucosyltransferase (UGT) modification, such as methylation, different anthocyanins are generated and collected into vacuoles for storage under the action of glutathione S-transferase (GST) transporters [71]. Finally, colorless dihydroflavanols are converted into colored anthocyanins by anthocyanin synthase ANS [67]. After modification, such as glycation (Flavonoid 3-glucosyltransferase [UFGT]) and other UGT [70] sequestered into vacuoles for storage under the action of Glutathione S-transferase (GST) transporters [71].

## 4. Transcriptional Regulation Mechanism of Anthocyanins by MYB-TFs

R2R3-MYBs and R3-MYBs play crucial roles as transcriptional regulators in the production of anthocyanins. The recent view of regulation of anthocyanin synthesis includes control by single TFs, as well as structured regulation by an anthocyanin biosynthesis network [20,72,73] in which the ternary MBW complex of transcription factors, involving a variety of interactions and mechanisms, plays a key role in anthocyanin and proanthocyanin accumulation and inhibition. This network promotes the synthesis of anthocyanin through positive regulation and inhibits or reduces anthocyanin accumulation by negative regulation [74,75,76]. Gene homologs related to anthocyanin biosynthesis regulated by MYB-TFs were found by sequence searches and a proposed evolutionary tree was constructed (Table 1 and Figure 3).

### 4.1. Mechanisms of Positive Regulation

Most of the genes with positive regulatory effects on anthocyanin biosynthesis encoded R2R3-MYB factor. Sequence motifs encoding three potential alpha helixes are found in each incomplete repeated (R1, R2, R3), of which the second and third helixes could form a helix-turn-helix (HTH) structure [110]. Structural studies of the MYB DNA-binding domain interacting with a double DNA strand reveal that the third helices of both the R2 and R3 repeats are considered recognition helices, which bind directly to specific *cis*-acting regulatory DNA elements [110]. The third helix of R2 and R3 repeating sequences is the key to modulating the binding affinity of DNA, and the variation of their repeating sequences will affect the specific binding of R2R3-MYB to DNA [111,112]. The Leu→Glu substitution in the third helix of the R2 repeat of MYB protein P1 in maize causes a loss of DNA binding [113]. The DNA binding domain is conserved within the separate classes of TF and contain *trans*-acting motifs that bind to the *cis*-acting sequences of target gene promoters. TFs are identified by their distinct DNA binding domains [114]. Therefore, the MYB-TF is able to bind directly to the promoter of the gene key enzyme gene for anthocyanin biosynthesis, affecting the final enrichment of anthocyanin. A database of nucleotide sequence motif of cis-acting elements in plants was discovered using PLACE (http://www.dna.affrc.go.jp/htdocs/PLACE/) (accessed on 1 February 2021). Flavonoid-related MYB binding elements were identified, including MYB26PS [115], MYB core [116], MYBPLANT [43], and MYBPZM [117]. In tomato (*S. lycopersicum*), *SlMYB75* is able to directly bind to the MYBPLANT and MYBPZM -regulatory elements and to activate the promoters of the structure genes, promotes anthocyanin accumulation and enhances volatile aroma production in tomato fruits [118]. The MYB-TFs bind indirectly to promoter-specific cis-acting elements by interacting with other DNA-binding proteins [119]. The *MYB10* can regulate skin color of apples varieties, which has been shown to auto-activate its own expression by binding a cis-element in its own promoter, then activating the expression of anthocyanin structural genes [114]. Synthetic promoter engineering may find targeted enhancers more precisely through the presence of cis-regulatory elements.

#### 4.1.1. Expression Patterns of MYB Activators

Expression patterns of MYB activators vary in vegetative and reproductive plant tissues. *MdMYB110a* controls ‘red meat’ coloring in red heart apples, and this pigment is only expressed in the fruit cortex late in fruit development [12], which indicates that MYB-TFs have different targeting effects on anthocyanin pigment deposition in different tissues. *PhAN2* is responsible for flower pigmentation in *Petunia hybrida*. The apparent orthologs in (*Lilium davidii*)*, LdMYB12* and *LdMYB6* showed high amino acid sequence similarity with PhAN2. *LdMYB12* is involved in anthocyanin accumulation in sepals and petals (perianth) sheets, filaments and stigma, while *LdMYB6* is mainly responsible for perianth sheet spots and dominant expression in flowers and bulbs [120]. Therefore, anthocyanin deposition in different parts of the plant is influenced by different gene activators. In kyoho grape (*Vitis labruscana*), *MybA* gene expression increases strongly with the commencement of coloring and berry softening and is detected only in berry skin and flesh. When *MYBA* was introduced into somatic embryo of grape, led to reddish-purple spots in non-colored embryos [70]. In nectarine (*P. persica*), *PpMYB10.1*, *PpMYB10.4*, and *PpMYB9*/*PpMYB10* were positive regulators of anthocyanin accumulation in fruits, leaves and flowers, respectively [25]. The expression level brought about by activators also depends on the predetermined level of anthocyanin in different genotypes [109,121]. In *white clover (Trifolium repens)*, the pigmentation of red stripes on the leaves is regulated by the R2R3-MYB family and anthocyanins are found at the R and V pigmentation sites [107]. Moreover, the *atv (atroviolacium)*, *Aft (Aubergine fruit) and Abg (Aubergine)* loci of tomato can also promote the pigmentation resulting from anthocyanins in fruits. After binding with Aft or *Abg* loci, *atv* loci can significantly increase the anthocyanin content in cultivated tomato fruit [104,122]. But in *Arabidopsis*, activation tagging induced to identification of a bright-purple mutant (*pap1-D*) in which overexpression of a MYB factor led to massive accumulation of anthocyanins in the entire plant [123,124]. By linking with enhancer sequences, indicates that activation tagging can be used to overcome the stringent genetic controls regulating the developmental accumulation of specific natural products [123]. According to this situation, anthocyanin color vision may be used to isolate candidate regulatory genes and express easily screened marker genes under the control of promoters from genes encode enzymes involved in the biosynthesis of natural products of interest [123].

#### 4.1.2. Regulation of the Structural Gene Network

MYB-TFs regulate anthocyanin structural genes that then activate anthocyanin biosynthesis. In the R2R3-MYB-TFs, the highly conserved N-terminus binds to the bHLH TF and WD40 repeat (WDR) to form the MYB-bHLH-WD40 (MBW) complex. The accepted view is that the MYB protein determines the specific activation of target genes [30]. In *painter’s pallet (Anthurium andraeanum)*, *AaMYB2* is considered to be the R locus ortholog that controls corolla color inheritance. As a potential target of coding TFs, the locus is involved in the joint expression of *AaCHS*, *AaF3H* and *AaANS*. *AaMYB2* is highly expressed in the spathes (sheathing bract) of red, pink and purple cultivars but hardly detected in the spathes of white and green cultivars [125]. As a late synthesis gene of anthocyanin biosynthesis, *anthocyanidin synthase (ANS)* is an important target gene. In a study of binding specificity of the kiwifruit (*Actinidia chinensis*) AcMYB75 protein, the promoter of *ANS* was identified in a yeast single hybrid experiment [78]. When the MYB-TF bound to the promoters of structural genes in anthocyanin biosynthesis pathway, the expression of structural genes was increased so that the entire downstream network of anthocyanin biosynthesis was activated. This illustrates that the expression of structural genes in the anthocyanin synthesis pathway was positively correlated with the accumulation of anthocyanin [126]. Activators can combine with other TFs to form complexes that participate in the expression of anthocyanin structural genes and stimulate the anthocyanin synthesis system. In addition to direct action on target gene sites, expression of anthocyanin MYB-TFs may be controlled by binding with other TFs. The R3 domain of positive regulatory factor shows a conserved bHLH binding motif, [D/E]Lx_2_[R/K]x_3_Lx_6_Lx_3_R (Appendix A). R2R3-MYBs usually interact with *bHLH1* and WD40 to create an MBW activation complex, thereby augmenting the expression of *bHLH2* and the structural genes to promote the accumulation of anthocyanin [44,106,112]. In tomato (*S. lycopersicum*)), the Aft (Anthocyanin fruit: MYB) protein interacts with SlJAF13 (bHLH) and SlAN11 (WDR) to form an MBW-activated complex and activate the expression of SlAN1 (bHLH). With the formation of a core MBW-activated complex of SlAN1, Aft and SlAN11, the expression of the *SlAN1* gene and most of the anthocyanin structural genes were activated, and anthocyanin pigments were enhanced in fruits [127]. *MrMYB1* from Chinese bayberry (*Myricarubra Chinese*) when genetically transformed into *N. tabacum*, the activation of the *NtDFR* promoter by *MrMYB1* depends on the expression of bHLH TFs. When *MrMYB1* is co-expressed with bHLH, the formation of anthocyanins is promoted [96]. Similarly, the heterologous expression of *MYBA* from tobacco can independently regulate *AtDFR* in grape hyacinth (*Muscari armeniacum*), evidenced by strong purple anthocyanin pigment accumulated in leaves, petals, anthers and calyx, while *MaAN2* relies on *MabHLH1* to control the expression of *AtDFR* to promote the synthesis of anthocyanin [128]. In orchids (*Phalaenopsis equestris*), *PeMYB2*, *PeMYB11*, and *PeMYB12* can independently activate *PeDFR* expression, when the presence of *PeBHLH1* the red pigmentation of butterfly orchids can be increased on this basis [129]. Similar results were found in peach (*P. persica*) [130], which indicates that the bHLH TF does not act as a restrictive regulator of anthocyanin accumulation in some plants but acts as a cofactor that promotes the biosynthesis of anthocyanin. In tree peony *(Paeonia × suffruticosa)*, *PsMYB12* interacts with bHLH and WD40 in the protein complex to directly activate the expression of *PsCHS*, showing pigmentation in the form of a blotch, while the loss of *CHS* activity leads to albino flowers [75,131,132]. The activity of the *MdDFR* promoter was enhanced significantly when *MdMYB10* was co-transformed with apple bHLHs [133]. Meanwhile, *Pr-D* (*Pr*: *Purple*, encoded a R2R3-MYB-TF) overexpressed in *wild cabbage (Brassica oleracea)* it activates bHLH TFs acting on *BoF3H*, *BoDFR*, and *BoANS* [36,134]. Therefore, many lines of evidence support the conclusion that MYB activators control the anthocyanin synthesis pathway by regulating the expression of anthocyanin structural and the control of anthocyanin accumulation.

### 4.2. Negative Regulatory Mechanism

MYB-TFs can also repress anthocyanin biosynthesis. In the transcriptional structure of MYB factors, there are not only transcriptional activation regions but also transcriptional inhibition regions. The first negative regulatory MYB gene (*AmMYB308*) was identified in snapdragon *(Antirrhinum majus)* [43]. Overexpression of *AmMYB308* in older leaves of tobacco resulted in the virtual absence of the soluble phenylpropanoid derivatives including two flavonoids. Subsequently, MYB-TFs that reduced the accumulation of anthocyanins in various plant tissues or organs were found in species, such as *peach, PpMYB19 (P. persica), poplar, PtrMYB182 (P. tremula* × *Populus), daffodil, NtMYB2 (Narcissus tazetta), apple, MdMYB6 (M. domestica), grape, VVMYBC2-L1 and VVMYBC2-L3 (Vitis vinifera), and strawberry, FaMYB1 (Fragaria × ananassa)* [11,44,47,98,101,110]. MYB-TFs are known to have one or more domains near in the C-terminus, which can induce transcription through specific binding to target sites [84,135]. Based on the clustering of the conserved motif (a fragment of a protein that is unable to interact) (in Figure 4), it was confirmed that the conserved motif was found in most plants. The C1 motif may have an assumed activation function but the role of C1 motif has not been characterized yet [11,135]. The C2 motif is also known as the EAR (for ethylene-responsive factor [ERF]-associated amphiphilic repression) motif, and it is the most common C-terminal restraint motif. The C2 region with the core EAR motif is considered to be the predominant transcriptional repression motif in plants [136,137,138]. C3 and C4 motifs exist in a small segment of the R2R3-MYB structural domain, and the motif inhibits benzene propane biosynthesis [84,139,140]. The C5 (TLLLFR) motif in the AtMYB-like of R3-MYB, participates in the formation of the MBW complex in bHLHs. The fourth subgroup of R2R3-MYB is defined by the existence of C1, C2, and C3 motifs [98,110]. In the regulation of flavonoid biosynthesis, MYB repressors mainly exist in the DNA binding domain with two repeats of R2R3-MYB and a single repeat of R3-MYB.

#### 4.2.1. R2R3-MYB Repressors

Whereas R2R3-MYBs can act as activators or repressors of anthocyanin biosynthesis, with other TF binding transcription repressors act through a variety of mechanisms [11,141]. In a R2R3-MYB of *Brassica rapa*, *BrMYB4* can inhibit the expression of *BrC4H* directly to the promoter, thus inhibiting anthocyanin accumulation [85]. The poplar MYB, *PtrMYB57* (*P. tremula* × *Populus*) interacts with *bHLH131* (bHLH) and *PtrTTG1* (WDR) to form an MBW complex, which competes with the activation complex to bind the promoter and inhibits the of activate the anthocyanin regulatory network [142]. *AtMYB7* (*A. thaliana*) and *FaMYB1* (*Fragaria* × *ananassa*) were identified and found to downregulate anthocyanin biosynthesis [37,143]. The proposed phylogeny (Figure 4), for R2R3-MYB-TF, suggests two ways to regulate flavonoid metabolism: AtMYB4-like (direct binding) [74,92,99] and FaMYB1-like (combined with other proteins) [44,74,104].

##### AtMYB4-Like Regulation

AtMYB4-like regulation acts directly on the promoters of anthocyanin structural genes to inhibit their expression [74,92,99]. In the negative regulation of anthocyanins, each repressor usually contains a small repression domain. Each type of repression motif is conserved and contacts specific promoter targets resulting in inhibition of gene expression [144]. The C1 motif and EAR(C2) repressor motifs are contained in the C-terminus of the AtMYB4-like regulators. A small number of C3 and C4 motifs are also present [52,145,146]. For example, *NtMYB2* (daffodil, *N. tazetta*) contains three typical conserved motifs, C1, C2, and C3, at the C-terminus, which participate in the inhibition of anthocyanin deposition [99]. In apple, *MdMYB16* directly inhibits the expression of *MdUFGT* and *MdANS* via a C-terminal EAR motif [46], which inhibits the biosynthesis of anthocyanins. If the EAR motif at the *MdMYB16* C-terminus is removed, anthocyanins increase [99]. In the AtMYB4-like evolutionary branch, *VvMYB4a* reduces anthocyanin by inhibiting the expression of the *DFR*, *ANS* and UFGT genes [99]. *AtMYB7* of *Arabidopsis* also uses *DFR* and *UFGT* as early targets to inhibit flavonoid pathways [147]. Overexpression of the apple homolog *MdMYB6* in calli of apple red fruit also reduces the anthocyanin content. *MdMYB6* binds directly to the promoters of *MdANS* and *MdGSTF12*, thus inhibiting the synthesis of anthocyanin [86]. Inhibition of *ANS*, *DFR*, and *UFGT* is a common feature of all transgenic tobacco plants and other plants transformed with AtMYB4-like type repressors [46,74]. The AtMYB4-like repressor inhibits anthocyanin accumulation by binding to the promoter of the terminal structural gene of the anthocyanin pathway (in Figure 5) [11,74]. In *Arabidopsis*, this mechanism of action extended our understanding of the AtMYB4-like repressor explains at least in part the complexity of the negative regulation.

##### FaMYB1-Like Regulation

FaMYB1-like transcription factors bind to other proteins to repress the expression of core activating complexes or to compete with activator complexes to suppress the expression of structural genes by promoters [44,74,104]. Overexpression of *FaMYB1* in tobacco inhibited anthocyanin synthesis [11]. *FaMYB1* does not contain a functional activation domain and the expression of anthocyanin structural genes was not affected by the overexpression of *FaMYB1*. Therefore, *FaMYB1* inhibits anthocyanin synthesis by a different mechanism than the AtMYB4-like class. Yeast two-hybrid binding indicated that *FaMYB1* interacts with two bHLH proteins (*JAF13* and *AN1*), which can act as repressors of anthocyanin biosynthesis when interacting with the core MBW activated complex [44,74,104]. Similar effects were confirmed observed in petunia *PhMYB27* (*P. hybrida*), grape *VvMYBC2* (*V. vinifera*) and *VvMYBC2-L3*, apple *MdMYB15L* (*M. domestica*), and poplar *PtrMYB182* (*P. tremula*) [42,46,47]. This regulatory mechanism is related to the action of the conserved C-terminal motif of R2R3-MYB. In the conserved bHLH region in the R3 domain of *PtrMYB182*, this MYB evolutionary branch has neither a zinc finger structure nor a C4 motif, but it does contain conserved C1 and C2 motifs and bHLH binding regions [47]. The absence of the bHLH binding site apparently eliminates the repressor activity of *PtrMYB182*, indicating that its activity depends on the interaction with the bHLH domain. The same result was obtained with *PhMYB27* and *VVMYB4-like* regulators [47,146]. The C2 motif is associated with the anthocyanin repressor *PhMYB27*, but the addition of C2 motif mutations in *PtrMYB182* does not reduce its repressor activity, indicating that the activity of such repressors is independent of the C2 motif [144,149] and suggesting that other yet unknown regulatory mechanisms are important. The characteristic action of FaMYB1-like factors is that they cannot directly bind to the promoter of the target gene but that they are integrated into the MBW complex interfere with the correct assembly (in Figure 5) [47]. According to this proposed regulatory mechanism, the genes targeted by FaMYB1-like repressors are the same as those targeted by the anthocyanin MBW activation complex, but that the role of the complex is changed change the complex activity and transform from activation to inhibition [47,74].

#### 4.2.2. R3-MYB Repressors

The MYB family also includes R3-MYB repressors containing a single MYB R3 repeat. In the R3-MYB-TFs, there are two domain types involved in anthocyanin inhibition. In one type, there is only one R3 domain, which does not directly contact DNA and has a conserved motif, [D/E]Lx_2_[R/K]x_3_Lx_6_Lx_3_R, that participates in interactions with BHLH-TFs, namely, AtCPC-like (*cpc*: caprice) activity [83,92,111]. The other domain contains a part of R2 and a complete R3 domain, which also interacts with the bHLH TF to inhibit anthocyanin biosynthesis, specifically AtMYBL1-like activity [106].

##### The AtCPC-Like Regulation

The AtCPC-like regulator is a single MYB-type protein that acts as a repressor in anthocyanin biosynthesis [150]. Six gene models of the R3-MYB-TFs were found in the *Arabidopsis* genome: *CPC*, *TRY*, *TCL*, *ETC1*, *ETC2* and *ETC3*. These genes play an important role in regulating root hair differentiation and trichome germination [151]. *CPC* and *TRY* (a negative regulator of trichome initiation in *Arabidopsis*) both inhibit the synthesis of anthocyanin. *CPC* is not conserved motif that can act as a direct activator of transcription [151], but CPC retains the binding motif of bHLH and exerts passive inhibition by competing with the R2R3-MYB activator for bHLH [74]. The homologs *PhMYBx* of petunia and *SlMYBATV* of tomato also use this regulatory mechanism to regulate anthocyanins. In petunia, *PhMYBx* is able to bind the bHLH component from the MBW complex, thereby inhibiting anthocyanin biosynthesis [44,152]. Yeast two-hybrid and transient expression in tobacco both showed that the GtMYB1R1 ((from Japanese gentian; *Gentiana triflora*) and GtMYB1R9 proteins interact with GtbHLH1 proteins and control anthocyanin accumulation by binding to *GtDFR* promoters to inhibit their expression. Overexpression of *GtMYB1R1* and *GtMYB1R9* significantly reduces the anthocyanin pigmentation in tobacco petals, resulting in a white-flower phenotype [90]. Similarly, in poplar mutants, R3-MYB repressors block anthocyanin accumulation together with bHLH binding to exert passive inhibition or binding to the MBW-activated complex to trigger active inhibition (Figure 5). This competitive repressor eventually depletes the required TF and is known as a “squeeze” [44,153]. Many mechanisms have been proposed for the action of this repressor, in which the activator activates the repressor, the repressor suppresses the activator, and the activator self-activates [44,154]. This model is consistent with the anthocyanin-related MBW regulatory network of petunia [148]: *PhAN1* (ANTHOCYANIN1) can activate the regulatory mechanism of *PhMYBx* (R3-repressor); *PhAN1* activates its own expression; and *PhMYBx*, in turn, inhibits *PhAN1* activity. The competitiveness of AtCPC-like repressors is affected by binding protein affinity, suggesting that regulation of binding proteins might alter anthocyanin inhibition. The outcome of such a mechanism can be homeostasis.

##### The AtMYBL1-Like Regulation

In contrast to the AtCPC-like, the AtMYBL1-like factor has a conserved EAR or “TLLLFR” motif of anthocyanin synthesis [150]. The R2R3-MYB repressor contains the structural domains at its C-terminus required for repressor activity [43,44]. In contrast, *PhMYB27*, as a negative regulator of R2R3-MYB, has a dual locking mechanism, which achieves repression by preventing the formation of the MBW complex or by competing with the activator of MBW to bind to the MBW-activated complex and convert it into an repressory complex. This repression requires a conserved EAR motif to achieve repression [44]. To date, researchers have found that in the R3 -MYB-TFs of an ortholog (*Lilium hybrid*), there is a C2 motif downstream of the MYB repeats of two genes, *LhR3MYB1* and *LhR3MYB2*, and the C2 motif in R2R3-MYBs can directly inhibit the expression of their target genes [92,104,114]. Transient expression of the two genes, *LhR3MYB1* and *LhR3MYB2*, showed that *LhR3MYB2* inhibits the expression of the GUS gene driven by the *DFR* promoter in *L. hybrid*, while *LhR3MYB1* does not. Whether the two can interact with the bHLH protein is not clear. However, *AtMYBL2* is also a R3-MYB-TF and does not have a C2 suppression motif but does have a conserved TLLLFR motif, which is functionally similar to the factor of the subtribe R2R3-MYB [44,155]. The repressor functions of *AtMYBL2* can be demonstrated by competing with the R2R3-MYB-TF, which activates anthocyanin synthesis with bHLH [84,150]. Unlike AtCPC-like regulation, *AtMYBL2* has a conserved repression motif. Similar results were observed in *IlMYBL1* (from the purple tube flower; *Iochroma loxense*), *AtMYBL2*, and *SmelMYBL1* (*eggplant*: *S. melongena L.)*) belonging to the same evolutionary branch, showing high similarity to the anthocyanin inhibition mechanism [90,92,106]. The homolog of *IlMYBL1* [92] in tomatoes strongly inhibits the accumulation of anthocyanin, leading to a deficiency of anthocyanin in fruits and flowers [155]. Overexpression of *SmelANT1* and *SmelAN2* results in a red phenotype in poplar leaves, while their co-expression with *SmelMYBL1* prevents anthocyanin accumulation, showing that *SmelMYBL1* inhibits the MBW complex by competing with the MYB activator for bHLH binding sites [106]. However, the inhibition of R3-MYB is rare, and the role of the conserved repression motif in this process deserves verification.

## 5. Environment Affects MYB Gene Regulation of Anthocyanin Biosynthesis

Environmental conditions affect plant growth and development. Plants respond to abiotic stress by production of protective metabolites. The regulation of anthocyanin biosynthesis by MYB factors is known to be affected by environmental stress or other external signals [103,156]. Genes associated with light stress or other abiotic stress responses (such as Ultraviolet-B light (UV-B) radiation, low temperature or drought) may exert strong effects. Under blue and far-red light, *AtMYB75* expression increased at low temperature while drought enhanced *AtMYB114* expression [37]. Environmentally induced mediation of MYB-TFs in anthocyanin biosynthesis may reveal fundamental features of the hierarchical genetic regulatory networks active during responses to environmental and developmental signals.

### 5.1. Light

Plant maturation requires appropriate light stimulation. UV-B is one environmental signal perceived by plants that affects the flavonoid pathway and influences the level of anthocyanins [157,158]. Apple trees were grown under spectral filters that altered transmission of solar UV light, ripening was delayed, fruit size decreased, and anthocyanin and flavonols were reduced [158]. Thotoreceptor scryptochrome 1 (CRY1) and phytochrome (PHY) can also induce the accumulation of anthocyanins [24]. Anthocyanin accumulates when the intensity of light is too high because anthocyanin protects photosynthetic organs [51,159]. Leucine Zip TF elongated hypocotyl 5 (HY5) is part of the light signaling pathway downstream of PHY, CRY and UV-B photoreceptors, which regulating the regulatory network of *PAP1* activates anthocyanin biosynthesis [24]. In *Arabidopsis*, *hy5* regulated *pap1* expression by directly combining G-box and ACE elements, while *pap1* acted on the promoters of EBGs and LBGs to induce the downstream network activation of anthocyanins [24,160]. *PAP1* expression was highly dependent on the presence of the HY5 TF, and this optical signaling pathway was affected by photosynthesis [24]. In the Asian pear (*Pyrus pyrifolia*), *PyHY5* was also determined to directly recognize and combine with *PyMYB10* and *PyWD40* to activate anthocyanin biosynthesis [103]. Photosensitivity plays an important role in the regulation of anthocyanins. In *Arabidopsis*, the blue light photoreceptor CRY1, red light photoreceptor phyB and far-red light photoreceptor phyA can promote anthocyanin accumulation under blue, red and far-red light, respectively [82]. When *Arabidopsis* was exposed to far-red light, phyA could promote anthocyanin accumulation by enhancing *AtMYB75* abundance and *AtDFR*, *AtLDOX* and *AtUFGT* expression [82]. To maintain the balance of anthocyanins, light signaling pathways downstream of constitutively photomorphogenic 1 (COP1: ubiquitin E3 ligase morphogenesis protein) TF suppress the accumulation of anthocyanins [161]. COP1 is found in the nucleus under dark conditions, where it targets photomorphogenetic activating TFs and mediates their degradation [162]. Apple *MdMYB1* is ubiquinated and degraded by *MdCOP1* dependent activity and fruit coloration in the dark is inhibited [162]. COP1 can act as a central switch of optical signal transduction, directly interacting with the photoreceptor and downstream target proteins [163,164]. CRY1-COP1-MYB75 and phyB-COP1-MYB75 signal transduction modules inhibit the accumulation of anthocyanins under blue and red light [82]. COP1 and HY5 anthocyanins are regulatory factors under the control of light. The HY5 TF is negatively regulated by COP1 [165], because COP1 can target the degradation of many TFs [164]. The genomic expression of anthocyanins may be regulated by the control of COP1 activity [164] to maintain an appropriate balance [161].

### 5.2. Temperature

Changes in temperature affect the regulation of anthocyanin biosynthesis by MYB-TFs. Strong light affects the accumulation of reactive oxygen species (ROS), a major component of oxidative stress. To ameliorate this stress, plants produce antioxidants such as ascorbic acid and a diverse group of polyphenols [28,166]. The protective effect of ascorbic acid in plants is inhibited at low temperatures, leading to the synthesis of additional antioxidants and the activation of anthocyanin biosynthesis [28,167]. Low temperature increases the accumulation of anthocyanin, which has been confirmed in apple (*M. domestica*) [168], rapeseed (*Brassica rapa*) [169] and eggplant (*Solanum melongena*) [39]. The expression of *CHS1* and *MYB1* at 6 °C was upregulated in gerbera daisy (*Gerbera jamesonii*), and the anthocyanin content was increased [143]. Similarly, in apples, the combination of *MdbHLH3* and *MdMYB1* increased the expression of *MdDFR* and *MdUFGT*, and low temperature improved the binding ability between bHLH and MYB and thereby promoting the biosynthesis of anthocyanin [39,168]. For anthocyanin activation by cold stress, C repeated binding factor (CBF) was found to play a central regulatory role in eggplant under cold stress [39]. A yeast two-hybrid experiment found that *SmCBF2* and *SmCBF3* upregulated the expression of *SmCHS* and *SmDFR* through the physical action of *SmMYB113* [168]. When *SmTT8* (a bHLH-TF partner of *SmMYB113*) were expressed together with *SmCBFs* and *SmMYB113*, the anthocyanin content was significantly higher than when *SmMYB113* was expressed alone [39]. Therefore, *SmMYB113* is the bridge for the mutual interaction between *SmCBFs* and *SmTT8* [39]. Two bZIP-TFs, *HY5* and a *HY5* homolog *HYH*, induced anthocyanin accumulation at low temperature in *Arabidopsis* [170]. These two factors require light to upregulate *DFR* at low temperature. They can also activate the downstream network of anthocyanins.

### 5.3. Other Factors Regulation Anthocyanin

Other environmental factors also affect the regulation of anthocyanin biosynthesis by MYB-TFs. Not only do light factors and low temperature regulate anthocyanin accumulation, but phytohormones, drought and high sucrose stress also induce MYBs that activate the regulatory network of anthocyanin biosynthesis. Drought stress that induces excessive ROS, can damage the cellular structure and metabolism of plants. To alleviate the damage caused by ROS stress, the antioxidant capacity is improved by activating the regulatory network and biosynthesis of anthocyanins [138,171]. Under drought stress, HD-ZIP II TF HAT1 negatively regulates (inhibits) anthocyanin accumulation by binding to the MBW protein complex [138]. In addition, the phytohormone abscisic acid (ABA) activates *MYB75*, a possible anthocyanin accumulation-induced TF, by inhibiting the physical interaction between HAT1 and *MYB75* [138]. Similarly, *IbMYB1* from sweet potato (*Ipomoea batatas*) can activate the accumulation of anthocyanin, improve the antioxidant capacity and enhance resistance to drought stress. *IbMYB1* overexpressed in *Arabidopsis* can also increase the tolerance to osmotic stress [172]. While the external environment may regulate the activation of anthocyanins by MYB- TFs, internal environmental factors (conditions within the plant) such as phytohormones and pH may affect expression of the regulatory network of MYB-TFs. In the MBW complex, JAZ proteins (repressors of the jasmonate signaling cascade) can interact with bHLH and MYB members to inhibit MBW formation and consequently inhibit anthocyanin biosynthesis. To activate the anthocyanin synthesis pathway, the degradation of JAZ proteins through the synthesis of jasmonic acid (JA) reduces its inhibition of MBW [82]. Moreover, JA relies on the photoreceptor encoded by *phyA* to induce *MYB75* expression and to consequently activate *DFR*, *UFGT* and *LDOX* and to regulate anthocyanin enrichment when *Arabidopsis* is exposed to far-red light [82]. The color of anthocyanins is pH dependent and therefore the color observed in plant tissues is affected by the pH in the vacuole [173]. As described in the Introduction of this review, the major anthocyanins cyanidin and peonidin are red at low pH and blue at high pH. Cyanidin, peonidin and delphinidin are all blue at high pH [32]. Genes controlling blue color expression may reduce internal vacuolar acidity by transporting sodium ions outside the vacuoles, intensifying the color of the blue pigments [174]. In petunia (Petunia × *hybrida*), *PhPH4* is a member of the R2R3MYB family, encoding a MYB domain protein expressed in the red epidermis of petals. Mutation of *PhPH4* causes the color to turn blue, presumably due to an increase in the internal pH, as reflected by an increase in the pH of petal extracts [173]. Anthocyanin accumulates under acidic pH conditions. In *Malus Crabapple*, low pH promoted the expression of the *McANS* and *McUFGT* genes, which increased anthocyanins and the red color of the leaves. The color changed from red to green with increasing pH [85]. The mechanisms of anthocyanin regulation in the internal and external environment are complex and multifactorial.

## 6. Epigenetic Effects of MYB-TFs on Anthocyanin Synthesis

R2R3-MYB-TFs change the conformation and stability of proteins through posttranslational modifications such as methylation, phosphorylation and ubiquitination [128,175].

### 6.1. Effect of MYB-TF Methylation on Anthocyanins

DNA methylation has played an important role in genetic regulatory mechanisms such as transposon silencing and gene expression [176]. DNA methylation and demethylation are essential for normal eukaryotic development. DNA methylation has been found at redundant cytosine residues (CG, CHH and CHG), where H equals “not G” [177]. In the WEREWOLF (WER) (*AtMYB66*) structural model of *Arabidopsis* MYB-TFs [178], R2 in the R2R3-MYB protein was associated with redundant GC elements and R3 specifically recognized AAC elements [178,179]. The methylation of the DNA at the 5th position of cytosine (5 mC) and the 6th position of adenine (6 mA) in the AAC element blocked the interaction between *WER* and DNA so that the accumulation of anthocyanin in apple peel is inhibited. The methylation level in the promoter region of MYB-TFs is related to anthocyanin accumulation [180]. The ‘Royal Gala’ apple has red stripes on the peel, whereas ‘Honeycrisp’ has green stripes. The green stripes are positively correlated with increased anthocyanin content, which is correlated with increased major anthocyanin biosynthetic gene transcript levels including *MdMYB10*. The *MdMYB10* promoter is highly methylated and methylation enrichment is observed in the green stripes [180]. In pears, methylation of three cytosines in *PcMYB10*, inhibits the accumulation of anthocyanin [179]. The relationship of DNA methylation and TF binding remains complex and is not sufficient to explain the contradictory correlations. Further exploration is needed.

### 6.2. Phosphorylation and Ubiquitination

Phosphorylation and ubiquitination also affect the regulation of anthocyanin biosynthesis by MYB-TFs. In *Arabidopsis* the light-induced accumulation of anthocyanins is dependent on MYB75, which is functionally dependent on phosphorylation by MPK4, a multifunctional MAP kinase [51,181]. MPK4 is a mitogen-activated protein (MAP) and its interaction with MYB75 and phosphorylation, is required for MYB75 regulation. This interaction signifies an important role for a MAPK pathway in light signal transduction [51]. When MPK4 phosphorylates *MYB75*, it improves the stability of *MYB75* but does not affect the binding of *MYB75* to the anthocyanin target promoter [51]. Both R2R3MYBs and MPK4 can function as positive regulators of anthocyanin biosynthesis [181].

When plants are harmed by cold stress or osmotic pressure, they activate MPK4 kinase, suggesting that MPK4 plays an important role in the interaction of the different stress response pathways induced by these signals leading to the accumulation of anthocyanins [148,182]. COP1 is known to be a molecular switch that regulates plant development and inhibits anthocyanin accumulation downstream of the light photoreceptor. When plants are exposed to light, light inactivates COP1 [82]. COP1 contains an N-terminal WD40 domain and has a conserved ring finger domain in the ubiquitin protein ligase subclass [51]. Under dark conditions, the COP1/SPAE3 (Suppressor of phyA E3 ubiquitin ligase) ubiquitin ligase degradation, targets gene *MYB75* and inhibits anthocyanin synthesis [138]. This is contrary to the effect of MPK4 on the stability of *MYB75*, indicating that MPK4 and COP1 jointly regulate the accumulation of photoinduced anthocyanins. The mechanisms of phosphorylation and ubiquitination in anthocyanins are complex and merit further investigation.

## 7. Integrated Control Network

The role of MYB-TFs in the regulation of anthocyanin biosynthesis depends on multiple regulatory factors that operate in a hierarchical manner [44]. The expression level of activator *PhAN1* (from *Petunia*) is low and that of MYB repressor *PhMYBx* is high under normal growth (noninductive) conditions [74]. *MYBx* repressors inhibit the formation of the MBW complex by binding to bHLH proteins, preventing the activation of the target gene *DFR* [44]. When the mature flowers and fruits of *Petunia* are photoinduced, the light signaling pathway activates R2R3-MYBs and MBW (including bHLH1) to form an activation complex, inducing the expression of bHLH [103,106,160]. Consequent bHLH2 forms an MBW complex with the activator and WDR protein and acts on the promoter of the late acting anthocyanin structural gene *DFR* to regulate the downstream network of anthocyanin biosynthesis [160]. *PhMYBx* repressors are also activated by MBW activating complexes, competing with activators for target gene sites for feedback inhibition [44]. To promote the biosynthesis of anthocyanins, plants prevent the activation of repressors by activating complexes through the mechanisms induced by environmental conditions. bHLH2 enhances inhibition [44]. Studies investigating the control of *Arabidopsis* hairy cells also confirmed this regulatory mechanism. Several environmental factors play an important role in the regulation of anthocyanins by MYB-TFs (Figure 6). The regulation of anthocyanin species by various MYB-TFs, appear to demonstrate this environmental tendency because the bHLH factor can bridge interactive regulation between MYB activators and MYB repressors. The overexpression of *MdMYB16* in apple calli inhibits the synthesis of anthocyanin, and the overexpression of *MdbHLH33* in the same callus shows that the repression effect of *MdMYB16* on anthocyanin biosynthesis is weakened [46,74]. Different plants exhibit different functional zoning of bHLH factors and play different roles in modulating regulation. In summary, the biosynthetic pathway of anthocyanins is regulated by the environment, repressors and activators, and the interaction of the MBW complex and various regulatory factors to maintain the balance of anthocyanins in plants and ensure the stability of plant pigment production.

## 8. Future Prospects

MYB-TFs play an important role in the regulation of anthocyanin biosynthesis and accumulation. Environmental factors fine-tune the control of anthocyanin biosynthesis to maintain the normal plant growth and development. Elucidating the regulatory network of MYB-TFs and its effects on anthocyanins will help us to better understand the activity of MYBs and their interactions. The expression patterns of the presumed functional MYB-TF orthologs in different plants are not identical and this variation enriches the network of plant phenotypes and environmental responses. There are still questions about the activity and evolution of this extensive regulatory network, which deserve further in-depth description and analysis. Many features of these cis-acting elements on their target gene promoters and their role in regulation of the anthocyanin biosynthetic pathway remain unknown. Similarly, the mechanisms by which environmental factors affect the regulation of activators and repressors in the anthocyanin regulatory hierarchy are also not established. The mechanisms by which specific TFs such as R3-MYB-TFs affect anthocyanins are poorly understood. What is the strength of MYB activator that is needed to activate the repressor? How is the strength of the activators and repressors determined? Does the MBW protein complex regulating anthocyanins, bind to other TFs and MYB-TFs in the same way? Exploring the regulatory mechanisms of MYB-TFs could help us to more fully understand the mechanisms of anthocyanin biosynthesis and the specific features of the major interactions of hierarchical genetic regulatory networks in the growth, development and adaptation of higher plants.

As more is known about the mechanisms of regulation and targeting of MYB-TFs, the elements and domains of regulation increase in value as components in the genetic engineers toolbox for the construction of novel regulatory sequences in synthetic biology. History of molecular biology shows that the best understood molecules of present studies become the reagents and building blocks of the future biochemical technology. Anthocyanins have great value in agriculture, nutriceuticals, medicine and basic plant sciences. Therefore, the design of pigments in advanced breeding technology is likely to make use of the intricate mechanisms of MYB-TFs and their roles in the hierarchical genetic regulatory networks of transcriptional control.

## Figures and Tables

**Figure 1 ijms-22-03103-f001:**
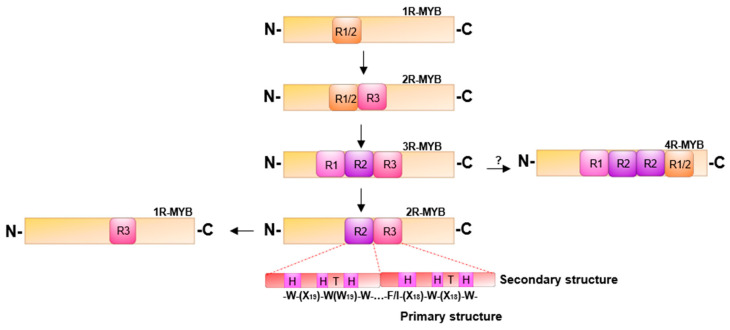
A proposed phylogenetic classification of MYB (*v-myb* avian myeloblastosis viral oncogene homolog) transcription factors (TFs) (arrows indicate the predicted direction of evolution [52,53]. Different plant protein types vary depending on the number of adjacent MYB duplications (R).

**Figure 2 ijms-22-03103-f002:**
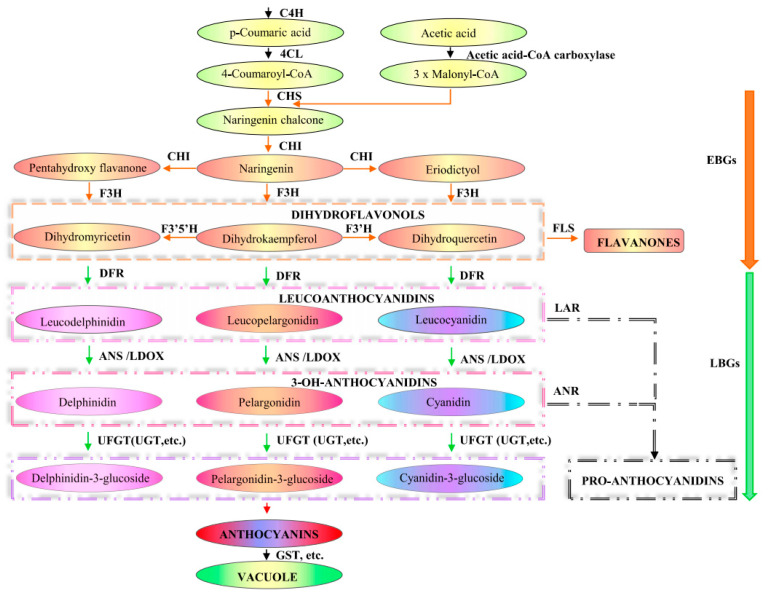
Model of the anthocyanin biosynthetic pathway. Enzymes are indicated in bold uppercase letters [23,26,65]. PAL, Phenylalanine ammonia lyase; C4H, cinnamate 4-hydroxylase; 4CL, Cinnamic acid-4-hydroxylase; CHS, chalcone synthase; CHI, chalcone isomerase; F3H, Flavanone 3-hydroxylase; F3′H, Flavonoid 3′-hydroxylase; F3′5′H, Flavonoid 3′,5′-hydroxylase; ANS, Anthocyanidin synthase; LDOX, Leucoanthocyanidin dioxygenase; UFGT, UDP flavonoid 3-O-glucosyltransferase; UGT, UDP -glucosyltransferase; GST, Glutathione S-transferase; FLS, Flavonol synthase; ANR, Anthocyanidin reductase; EBGs, early biosynthetic gene; LBGs, late biosynthetic gene.

**Figure 3 ijms-22-03103-f003:**
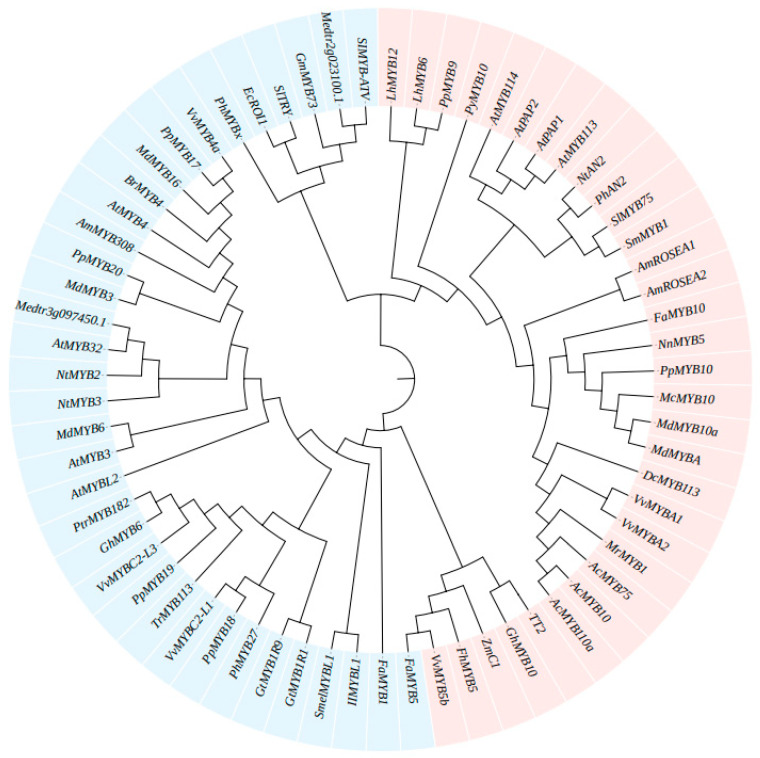
Phylogenetic tree of the regulation of anthocyanin biosynthesis by MYB genes. The tree was constructed based on the entire protein sequences using MEGA 6 software. Blue represents the antagonistic effects of MYB factors in the anthocyanin biosynthetic pathway, and red represents the MYBs as activators to promote the biosynthesis of anthocyanin.

**Figure 4 ijms-22-03103-f004:**
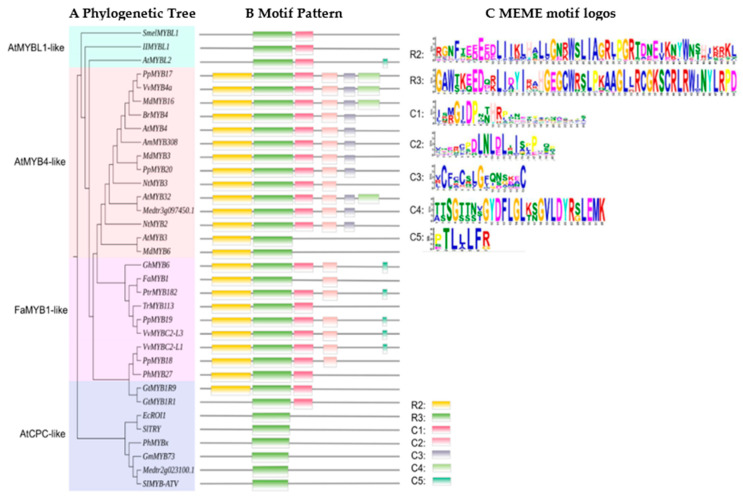
Phylogenetic relationships of conserved protein motifs in MYB transcription factor repressors related to anthocyanin biosynthesis [74]. (**A**) The phylogenetic tree was constructed based on the entire protein sequences using MEGA 6 software. Accession numbers are listed in Table 1. (**B**) The motif composition on every branch of the phylogenetic tree is represented by colored boxes. (**C**) The sequence for motifs R1-2, C1-5 identified by MEME.

**Figure 5 ijms-22-03103-f005:**
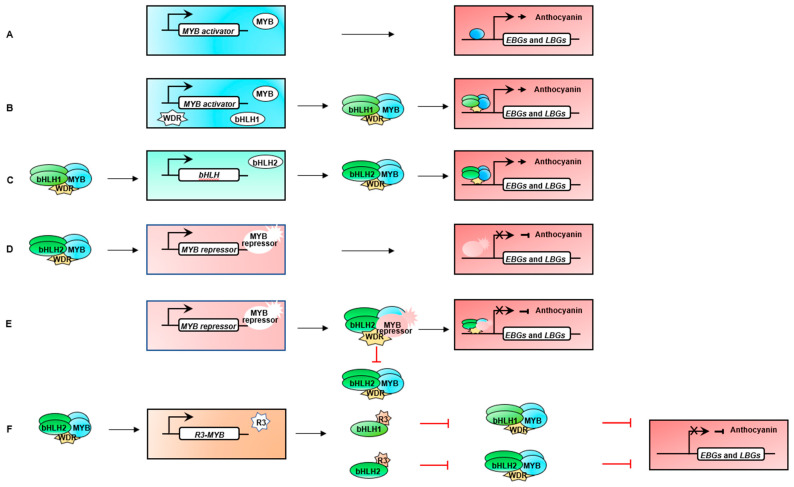
Proposed molecular mechanisms of MYB transcription factor in the regulation of expression of target genes [148]. (**A**) Anthocyanin synthesis is initiated in binding directly to target gene by the MYB activator. (**B**) The MYB activator, WD40 repeat (WDR), and bHLH1 proteins form an MBW activation complex that activated structural genes and promoted the accumulation of anthocyanin. (**C**) The MYB activator, WDR, and bHLH1 proteins form an MBW activation complex that activated the expression of bHLH2 and structural genes and promote the accumulation of anthocyanin. (**D**) MYB repressors are activated by the MBW complex, that binding to the promoter of the terminal structural gene in the anthocyanin biosynthesis pathway. (**E**) MYB repressors could inhibit anthocyanin biosynthesis by incorporating into the MBW activation complex as a corepressor or by binding to the promoter of target genes directly. (**F**) Feedback inhibition is provided by R3-MYB repressors; these are activated by the MBW complex and inhibit the formation of new MBW complexes by titrating bHLH factors.

**Figure 6 ijms-22-03103-f006:**
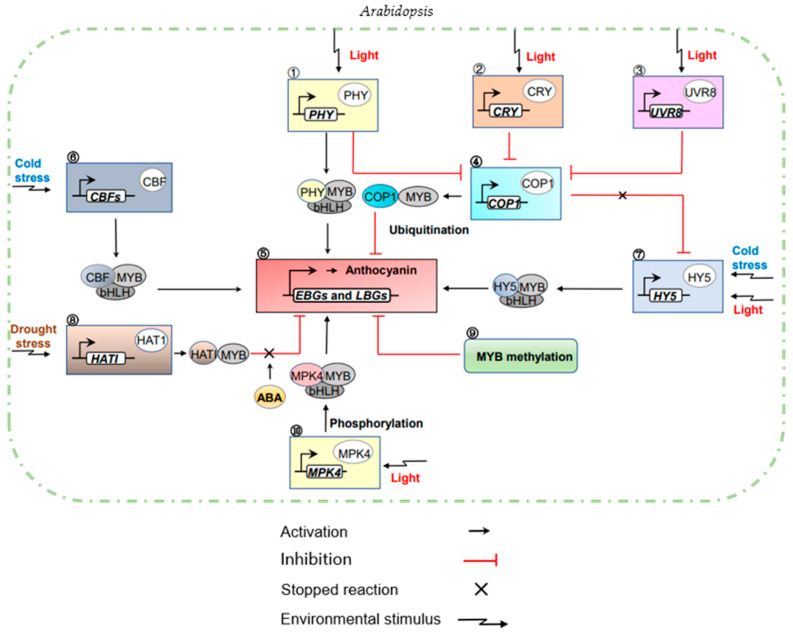
MYB transcription factors regulates the action network of anthocyanin in plants’ internal and external environment. Under light conditions, anthocyanin biosynthesis is initiated in developing leaves or developing flowers/fruits by PHY photoreceptors (**1**) binding directly to MYB and bHLH. At the same time, CRY (**2**) and UVR8 (**3**) photoreceptors could inhibit (**4**) COP1 factor. In the dark, COP1 degraded MYB transcription factors that reduce the expression of genes activating anthocyanin biosynthesis (**5**), ultimately inhibiting anthocyanin accumulation. Under cold stress, C repeated binding factor (CBF factor) (**6**) activated anthocyanin biosynthesis through physical interaction with MYB and bHLH. At the same time, HY5 factor (**7**) also induces anthocyanin accumulation so that the anthocyanin structural gene could be up-regulated under low temperature under light conditions to activate the downstream network of anthocyanins. Under drought conditions, HAT1 factor (**8**) negatively regulates anthocyanin accumulation by binding to MBW protein complexes, while in the presence of the plant hormone ABA, ABA could inhibit the physical interaction between HAT1 and *MYB75* and activate *MYB75* to induce anthocyanin accumulation. When the MYB target sequence was methylated (**9**), anthocyanin accumulation was also inhibited. Under light conditions, MPK4 (**10**) interacts with MYB and bHLH and phosphorylates MYB, then participating in the accumulation of light-induced anthocyanins.

**Table 1 ijms-22-03103-t001:** MYB-TFs in Regulation of Anthocyanins Biosynthesis in Plants.

Species	Gene	Type	Function	Accession	References
*Actinidia chinensis*	*AcMYB10*	R2R3	+	QGA78460	[77]
	*AcMYB75*	R2R3	+	APZ74276	[78]
	*AcMYB110a*	R2R3	+	AHY00342	[78]
*Antirrhinum majus*	*AmMYB308*	R2R3	−	ANTMA	[43]
	*AmRosea1*	R2R3	+	ABB83826	[79]
	*AmRosea2*	R2R3	+	ABB83827	[79]
*Arabidopsis thaliana*	*AtMYB3*	R2R3	−	BAA21618	[22]
	*AtMYB4*	R2R3	−	NP195574.1	[18]
	*AtMYB32*	R2R3	−	NP195225.1	[80]
	*AtPAP1*	R2R3	+	Q9FE25.1	[81]
	*AtPAP2*	R2R3	+	Q9ZTC3.1	[82]
	*AtMYB113*	R2R3	+	Q9FNV9.1	[82]
	*AtMYB114*	R2R3	+	Q9FNV8.1	[82]
	*AtMYB123(TT2)*	R2R3	+	Q9FJA2	[83]
	*AtMYBL2*	R3	−	NP177259	[84]
*Brassica rapasubsp. rapa*	*BrMYB4*	R2R3	−	ADZ98868.1	[85]
*Daucus carota*	*DcMYB113*	R2R3	+	QEE04281	[86]
*Erythranthe cardinalis*	*EcROI1*	R3	−	AGC66792.1	[83]
*Freesia hybrida*	*FhMYB5*	R2R3	+	QAX87835	[30]
*Fragaria* × *ananassa*	*FaMYB1*	R2R3	−	AF401220.1	[11]
	*FaMYB5*	R2R3	−	QIZ03072	[87]
	*FaMYB10*	R2R3	+	ABX79947	[83]
*Gossypium hirsutum*	*GhMYB6*	R2R3	−	AAN28286	[88]
	*GhMYB10*	R2R3	+	AF336282.1	[89]
*Gentiana triflora*	*GtMYB1R1*	R3	−	BAO51653.1	[90]
	*GtMYB1R9*	R3	−	BAO51654.1	[90]
*Glycine max*	*GmMYB73*	R3	−	ABH02868	[91]
*Iochroma loxense*	*IIMYBL1*	R3	−	ASR83104	[92]
*Lilium hybrid cultivar*	*LhMYB6*	R2R3	+	AZP55091	[93]
	*LhMYB12*	R2R3	+	BAO04194	[93]
*Malus hybrid cultivar*	*McMYB10*	R2R3	+	AFP89357	[94]
*Malus domestica*	*MdMYB1*	R2R3	+	ADE92935	[12]
	*MdMYB3*	R2R3	−	AEX08668	[76]
	*MdMYB6*	R2R3	−	AAZ20429.1	[86]
	*MdMYB10a*	R2R3	+	ABB84753.1	[36]
	*MdMYB16*	R2R3	−	ADL36756.1	[46]
	*MdMYBA*	R2R3	+	BAF80582	[31]
*Medicago truncatula*	*Medtr2g023100.1*	R3	−	KEH36848	[95]
	*Medtr3g097450.1*	R2R3	−	KEH35662	[95]
*Morella rubra*	*MrMYB1*	R2R3	+	ADG21957	[96]
*Nelumbo naceae*	*NnMYB5*	R2R3	+	ALU11263	[97]
*Narcissus tazetta subsp*	*NtMYB2*	R2R3	−	ATO58377.1	[98]
	*NtMYB3*	R2R3	−	AGO33166	[99]
	*NtAN2*	R2R3	+	ACO52472	[100]
*Peach (Prunus persica)*	*PpMYB9*	R2R3	+	ALO81019	[101]
	*PpMYB10*	R2R3	+	ADK73605	[102]
	*PpMYB17*	R2R3	−	ALO81020	[101]
	*PpMYB18*	R2R3	−	ALO81021	[101]
	*PpMYB19*	R2R3	−	ALO81022	[101]
	*PpMYB20*	R2R3	−	ALO81023	[87]
*Petunia hybrida*	*PhAN2*	R2R3	+	AAF66727	[45]
	*PhMYB27*	R2R3	−	AHX24372	[44]
	*PhMYBx*	R3	−	AHX24371.1	[6]
*Populus tremula* × *Populus tremuloides*	*PtrMYB182*	R2R3	−	AJI76863.1	[47]
*Pyrus pyrifolia*	*PyMYB10*	R2R3	+	ALN66630	[103]
*Solanum lycopersicum*	*SlTRY*	R3	−	AUG72363	[16]
	*SIMYB-ATV*	R3	−	NP001352307.1	[104]
	*SlMYB75*	R2R3	+	NP001265992	[105]
*Solanum melongena* L.	*SmelMYBL1*	R3	−	QJF74755	[106]
*Trifolium repens*	*TrMYB113*	R2R3	−	AMB27081.1	[107]
*Vaccinium corymbosum*	*VvMYB4a*	R2R3	−	ABL61515.1	[108]
	*VvMYB5b*	R2R3	+	NP001267854	[109]
*Vitis vinifera*	*VvMYBA1*	R2R3	+	AGH68552.1	[78]
	*VvMYBA2*	R2R3	+	BAD18978.1	[78]
	*VvMYBC2-L1*	R2R3	−	ABW34393	[110]
	*VvMYBC2-L3*	R2R3	−	AIP98385.1	[110]
*Zea mays*	*ZmC1*	R2R3	+	AAK81903	[45]

“+” represents that this MYB protein plays a positive regulatory role on anthocyanin accumulation in plants. “−” represents that this MYB protein plays a negative regulatory role on anthocyanin accumulation in plants.

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
