# Peer review of "MYB-Mediated Regulation of Anthocyanin Biosynthesis"

_ijms, 2021, doi:10.3390/ijms22063103_

Round 1
Reviewer 1 Report
The review is one of the so many review on anthocyanins and their regulation, however this ms include the epigenetic regulation which is for sure a novelty in respect to bibliography. The ms needs to be revised carefully fo english grammar, style (pay attention to italics where appropriate as welle as capital letters for acronyms) and scientific knowledge reported. Here fwe concerns:
Line 49: phenylpropane pathway instead phenylpropanoid
Lines 50-51 please clarify differences between monocot and dicots
Lines 52-57 shift before the paragraph on EBg and LBG
Line 61: are secondary metabolites of flavonoids modify with are a subclass of flavonoids
Line 75: clarify the differential regulation in dicots of EBG by single MTB and LBG by the MBW
Line 88: and they were change with when they were expressed…
Line 96 what you mean with information age ? here is a matter of increasing of genomic and transcriptomic data (omics era)
Line 103 Populus tomentosa should be in italics
Line 164 please rephrase is not clear: Anthocyanins are mediated by a variety of enzymes to regulate synthesis
Lines 253-257 rephrase it accordingly to tomato model and not the eggplant one! References to better describe this needs to be added. Please specify the identity of AFT.
Lines 301-303 rephrase it is not clear both for grammar and significance
Fig.1 is not supplementary thus should be delated from Supp. file
Author Response
Manuscript Number: ijms-1137418
Dear Professor,
Thank you for your letter and for the reviewers’ comments concerning our manuscript entitled “MYB-Mediated Regulation of Anthocyanin Biosynthesis”. Those comments are all valuable and very helpful for revising and improving our paper, as well as the important guiding significance to our researches. We have studied comments carefully and have made correction which we hope meet with approval. The main corrections in the paper and the responds to the reviewer’s comments are as flowing:
- Line 49: phenylpropane pathway instead phenylpropanoid
Response 1): line52-53 =>Anthocyanins typically accumulate in the vacuoles and their biosynthesis is mediated by many enzymes in the phenylpropanoid metabolic pathways.
- Lines 50-51 please clarify differences between monocot and dicots
- Lines 52-57 shift before the paragraph on EBG and LBG
Response 2,3): line57-66 =>In monocotyledon (maize, Zea mays [19], rice, Oryza sativa [20]), MYB-TFs regulate anthocyanin biosynthesis enzymes (such as CHS, F3H and DFR, LDOX, BAN/ANR, UFGT) together with other TFs [21]. In dicotyledon (Arabidopsis thaliana [22], apple, Malus domestica [12]), anthocyanin synthases are divided into two classes and the TFs are different [23]. The CHS, CHI, F3H and F3 'H genes are the early biosynthetic genes (EBGs) in the anthocyanin pathway. These genes bind directly and regulate by MYB-TFs [23]. DFR, LDOX, BAN/ANR, and UFGT, are the late biosynthesis genes (LBGs) [24-25], are regulated by the MBW-TF ternary protein complex of MYB-bHLH-WD40 that controls the MBW complex and the downstream accumulation of anthocyanins [26-29].
- Line 61: are secondary metabolites of flavonoids modify with are a subclass of flavonoids
Response 4): line80-82=> Anthocyanin are secondary metabolites of flavonoids modify with are a subclass of flavonoids, the Anthocyanin biosynthesis as well as that of other phenylpropanoids are regulated by MYB-TFs.
- Line 75: clarify the differential regulation in dicots of EBG by single MTB and LBG by the MBW
Response 5): line60-66=> In dicotyledon (Arabidopsis thaliana [22], apple, Malus domestica [12]), anthocyanin synthases are divided into two classes and the TFs are different [23]. The CHS, CHI, F3H and F3 'H genes are the early biosynthetic genes (EBGs) in the anthocyanin pathway. These genes bind directly and regulate by MYB-TFs [23]. DFR, LDOX, BAN/ANR, and UFGT, are the late biosynthesis genes (LBGs) [24-25], are regulated by the MBW-TF ternary protein complex of MYB-bHLH-WD40 that controls the MBW complex and the downstream accumulation of anthocyanins [26-29].
- Line 88: and they were change with when they were expressed…
Response 6): line101-106=> Subsequently, repressors of anthocyanin synthesis were found in strawberry FaMYB1 (Fragaria x ananassa), Arabidopsis AtMYB4, petunia PhMYBx (Petunia hybrida), peach PpMYB17-20 (P. persica), popular PtrMYB182 (Populus tremula x Populus), and apple MdMYB16 (M. domestica) [11, 44-48]. The anthocyanins biosynthesis pathway in the plant concerned is inhibited, resulting in a decrease in the level of anthocyanin in vivo when they were expressed.
- Line 96 what you mean with information age ? here is a matter of increasing of genomic and transcriptomic data (omics era)
Response 7): line114-115=> Predictive computational analysis has led to new uses for MYB-TFs by changing the amino acid sequence of the MYB domain to expand its regulatory ability.
- Line 103 Populus tomentosa should be initalics
Response 8): line118=> We had made correction according to the comments. The Populus tomentosa had been changed.
- Line 164 please rephrase is not clear: Anthocyanins are mediated by a variety of enzymes to regulate synthesis
Response 9): line184-188=> Anthocyanin biosynthesis is mediated and regulated by a variety of enzymes to regulate synthesis. The EBGs (CHS, CHI, F3H and F3‘H) and LBGs (DFR, LDOX, BAN/ANR, and UFGT) were initially categorized accordingly to their coordinate expression in response to environmental cues, such as light, at distinct developmental stages, in a species-dependent manner [23, 26-29].
- Lines 253-257 rephrase it accordingly to tomato model and not the eggplant one! References to better describe this needs to be added. Please specify the identity of AFT.
Response 10): line317-321=> In tomato (S. lycopersicum), the Aft (Anthocyanin fruit: MYB) protein interacts with SlJAF13 (bHLH) and SlAN11 (WDR) to form an MBW-activated complex and activate the expression of SlAN1 (bHLH). With the formation of a core MBW-activated complex of SlAN1, Aft and SlAN11, the expression of the SlAN1 gene and most of the anthocyanin structural genes were activated, and anthocyanin pigments were enhanced in fruits [129].
- Lines 284-292 rephrase it is not clear both for grammar and significance
Response 11): line317-321=> But in Arabidopsis, activation tagging induced to identification of a bright-purple mutant (pap1-D) in which overexpression of a MYB factor led to massive accumulation of anthocyanins in the entire plant [125, 126]. By linking with enhancer sequences, indicates that activation tagging can be used to overcome the stringent genetic controls regulating the developmental accumulation of specific natural products [125]. According to this situation, anthocyanin color vision may be used to isolate candidate regulatory genes and express easily screened marker genes under the control of promoters from genes encode enzymes involved in the biosynthesis of natural products of interest [125].
- 1 is not supplementary thus should be delated from Supp. file
Response 12): We had made correction according to the comments. The Supp. file had been exchanged, and unloaded a new file.

Reviewer 2 Report
Overall, this manuscript has a lot of information, but authors need to more succinctly describe and summarize it. Also if possible, it is better to state `take home message` at first, and then describe the examples in each part, so readers are able to catch easily the important points and remember their examples longer.
Sometimes, authors use the sentences with the similar meaning but different phrases, thus causing the content longer.
I don`t think it is appropriate to use just `***-like` as a title in some parts
Authors often use the word incorrectly. For example, a conserved EAR or `TLLLFR` inhibitor. EAR and TLLFR are not inhibitors but inhibitory motifs. These types of mistakes make readers confused.
Although authors described the sentences based on the refence, they use `will`. It is also inappropriate.
Author Response
Manuscript Number: ijms-1137418
Dear Professor,
Thank you for your letter and for the reviewers’ comments concerning our manuscript entitled “MYB-Mediated Regulation of Anthocyanin Biosynthesis”. Those comments are all valuable and very helpful for revising and improving our paper, as well as the important guiding significance to our researches. We have studied comments carefully and have made correction which we hope meet with approval. The main corrections in the paper and the responds to the reviewer’s comments are as flowing:
- Sometimes, authors use the sentences with the similar meaning but different phrases, thus causing the content longer.
Response 1): We had made correction according to the comments. I deleted many of the repetitive or confusing sentences and rewrote them.
such as: Line 114-117
With the advent of the information age, biological information has developed rapidly, and research on MYB TFs has become increasingly in-depth. Current studies have found that MYB TF-regulated anthocyanins can be used as visible selection markers for plant transformation.
=>
Predictive computational analysis has led to new uses for MYB-TFs by changing the amino acid sequence of the MYB domain to expand its regulatory ability.
such as: Line 184-187
We classified the CHS, CHI, F3H and F3 ‘H genes as early biosynthesis genes (EBGs) for the anthocyanin pathway. It is known that these genes are not directly controlled by any MBW compounds [54] but instead directly regulate factor combinations. At the same time, DFR, LDOX, BAN/ANR, and UFGT, called late biosynthesis genes (LBGs), could be MBW compounds used to identify specificity to promote the accumulation of anthocyanins [31, 55, 56].
=>
Anthocyanin biosynthesis is mediated and regulated by a variety of enzymes to regulate synthesis. The EBGs (CHS, CHI, F3H and F3‘H) and LBGs (DFR, LDOX, BAN/ANR, and UFGT) were initially categorized accordingly to their coordinate expression in response to environmental cues, such as light, at distinct developmental stages, in a species-dependent manner [23, 26-29].
such as: Line 270-273
Expression profiles of PeMYBs in Phalaenopsis ssp. showed that PeMYB12, PeMYB1 and PeMYB12 controlled the red color in the sepals, petals and lip, respectively. During transient expression of Phalaenopsis ssp., PeMYB11 responded to red spots in the callus, while PeMYB12 contributed to complete coloration of the central leaves [103]. Given the expression pattern of the R2R3-MYB activator in plant tissues, this activator was positively correlated with anthocyanin accumulation [85].
=>
In kyoho grape (Vitis labruscana), MYBA gene expression increases strongly with the commencement of coloring and berry softening, and is detected only in berry skin and flesh. When MYBA was introduced into somatic embryo of grape, led to reddish-purple spots in non-colored embryos [122]
such as: Line 408-409
However, a new study found that in Arabidopsis, MYB4 does not combine with the CHS or DFR promoter; rather, arogenate dehydratase 6 (ADT6) is the target that directly adjusts to the biological metabolism of phenylalanine. Moreover, the MBW com-plex and early biosynthesis genes (such as CHS) are indirectly inhibited by interfering with the transcriptional activity of the MBW complex [64]. This mechanism of action ex-tended our understanding of the AtMYB4-like repressor and explained explains at least in part the complexity of the negative regulatory mechanism, which deserves further study regulation.
=>
In Arabidopsis, this mechanism of action extended our understanding of the AtMYB4-like repressor explains at least in part the complexity of the negative regulation.
such as: Line 408-409
Plants respond to abiotic stress by altering the expression of genes needed to produce protective metabolites. Environmental conditions play an important role in the process of plant growth and development. In recent years, studies on the regulatory mechanisms of MYB factors on anthocyanins have shown that environmental stress or other signals have different responses to the activation of anthocyanins [86, 133].
=>
Environmental conditions affect plant growth and development. Plants respond to abiotic stress by production of protective metabolites. The regulation of anthocyanin biosynthesis by MYB factors is known to be affected by environmental stress or other external signals [103, 158].
- I don`t think it is appropriate to use just `***-like` as a title in some parts
Response 2): We had made correction according to the comments. The titles are changed “****-like regulation”.
- Authors often use the word incorrectly. For example, a conserved EAR or `TLLLFR` inhibitor. EAR and TLLLFR are not inhibitors but inhibitory motifs. These types of mistakes make readers confused.
Response 3): line360-363 =>The C2 motif is also known as the EAR (for ethylene-responsive factor [ERF]-associated amphiphilic repression) motif, and it is the most common C-terminal restraint motif. The C2 region with the core EAR motif is considered to be the predominant transcriptional repression motif in plants [138-140].
line 365-366 =>The C5 (TLLLFR) motif in the AtMYB-like of R3-MYB, participates in the formation of the MBW complex in bHLHs.
line 393-394 =>The C1 motif and EAR(C2) repressor motifs are contained in the C-terminus of the AtMYB4-like regulators. A small number of C3 and C4 motifs are also present [58, 147, 148].
line 397-399 =>In apple, MdMYB16 directly inhibits the expression of MdUFGT and MdANS via a C-terminal EAR motif [46], which inhibits the biosynthesis of anthocyanins. If the EAR motif at the MdMYB16 C-terminus is removed, anthocyanins increase [99].
Line 490-491 =>In contrast to the AtCPC-like, the AtMYBL1-like factor has a conserved EAR or “TLLLFR” motif of anthocyanin synthesis [152].
Line 496 =>This repression requires a conserved EAR motif to achieve repression [44].
Line 503-505 =>However, AtMYBL2 is also a R3-MYB-TF and does not have a C2 suppression motif but does have a conserved TLLLFR motif, which is functionally similar to the factor of the subtribe R2R3-MYB [44, 157].
- Although authors described the sentences based on the refence, they use `will`. It is also inappropriate.
Response 4): line 72-76=>Meanwhile, anthocyanins are considered beneficial for human health; presumably as antioxidants that reduce the abundance of free radicals (reactive oxygen species, ROS), which may delay aging, ameliorate cardiovascular and neurogenerative disease, as well as modulating gut microbiota [7, 33], however it has been argued that direct evidence of the benefits of dietary supplements are lacking [34].
Line 97-99=>Variation in relative abundance and specificity of anthocyanins represents one of the most common evolutionary changes in flower color. MYB-TFs may also act as antagonistic repression of anthocyanin enrichment [43].
line 289-291=>According to this situation, anthocyanin color vision may be used to isolate candidate regulatory genes and express easily screened marker genes under the control of promoters from genes encode enzymes involved in the biosynthesis of natural products of interest [125].
Line 358-359=>The C1 motif may have an assumed activation function but the role of C1 motif has not been characterized yet [11, 137].
Line 562-563=>The genomic expression of anthocyanins may be regulated by the control of COP1 activity [166] to maintain an appropriate balance [163].
line 615-617=>Genes controlling blue color expression may reduce internal vacuolar acidity by transporting sodium ions outside the vacuoles, intensifying the color of the blue pigments [176].

Round 2
Reviewer 1 Report
The ms has been improved according to reviewer's suggestions and it can be now accepted for publication
Reviewer 2 Report
I think authors have revised their writing as much as they can.
Overall, I think this paper could be suitable for publication.